# $CO_2$ exposure drives a rapid pH response in live adult *Drosophila*

**Sandra G. Zimmerman**(ID)*, **Celeste A. Berg**(ID)*

Department of Genome Sciences, University of Washington, Seattle, Washington, United States of America

* caberg@uw.edu (CAB); sgzimmerm@gmail.com (SGZ)

**Data Availability Statement:** All relevant data are within the manuscript and its Supporting information files.

**Funding:** This study was supported by National Institutes of Health, R01 GM079433, https://grantome.com/grant/NIH/R01-GM079433-01A2.

## Abstract

$CO_2$ anesthesia is the most common method for immobilizing *Drosophila* for research purposes. But $CO_2$ exposure has consequences—it can impact fertility, behavior, morphogenesis, and cytoskeletal dynamics. In this respect, *Drosophila* is an outstanding model for studying the impact of $CO_2$ exposure on tissues. In this study we explored the response of intracellular pH ($pH_i$) to a one-minute $CO_2$ pulse using a genetically encoded, ubiquitously expressed pH sensor, tpHusion, to monitor $pH_i$ within a live, intact, whole fly. We compared wild-type flies to flies lacking Imaginal disc growth factors (Idgfs), which are chitinase-like proteins that facilitate developmental processes and the innate immune response. Morphogenetic and cytoskeletal defects in *Idgf-null* flies are enhanced after $CO_2$ exposure. We found that $pH_i$ drops sharply within seconds of the beginning of a $CO_2$ pulse and recovers over several minutes. The initial profile was nearly identical in control and *Idgf-null* flies but diverged as the $pH_i$ returned to normal. This study demonstrates the feasibility of monitoring pH in live adult *Drosophila*. Studies exploring pH homeostasis are important for understanding human pathologies associated with pH dysregulation.

## Introduction

*Drosophila* researchers routinely immobilize flies with $CO_2$ without regard to the physiological consequences. $CO_2$ anesthesia, however, can have adverse effects: it impairs fertility, suppresses the immune system, and it can negatively impact climbing and flight behavior for hours or days [1, 2]. We demonstrated that $CO_2$ exposure enhances specific morphogenetic defects in flies with null mutations in *Imaginal disc growth factors* (*Idgfs*), a family of six genes that encode *Drosophila* chitinase-like proteins with orthologs in humans [3]. In addition, $CO_2$ exposure induces a loss of cortical actin during oogenesis and embryonic development in both wild-type and *Idgf-null* flies [3]. Wild-type flies proceed to develop normally, but *Idgf-null* flies do not, suggesting a possible role for *Idgf*s in protecting against adverse effects of $CO_2$. The mechanisms for how Idgfs mediate this protection are unknown.

How does elevated $CO_2$ induce these morphogenetic and cytoskeletal defects? One way could be by disrupting pH homeostasis ($CO_2$ reacts with water to form carbonic acid [4]). The actin cytoskeleton is extremely sensitive to pH, and pH changes can induce drastic changes in actin organization [reviewed in 5]. Dynamic remodeling of the cytoskeleton

The funders had no role in study design, data collection and analysis, decision to publish, or preparation of the manuscript.

**Competing interests:** The authors have declared that no competing interests exist.

drives cell protrusion, cell migration, and tissue morphogenesis and relies on the activity of pH-sensitive proteins such as cofilin [6–8], profilin [9], and Talin [10]. For example, actin polymerization at the leading edge of migrating cells increases with a pH$_i$ >7.2 in mammalian cells [11]; decreasing pH$_i$ or increasing extracellular pH (pH$_e$) inhibits cell migration [6, 12–14]. Cancer cells reverse their intracellular-to-extracellular pH gradient (pH$_e$ < pH$_i$) compared to normal cells (pH$_e$ > pH$_i$), thereby promoting cell protrusion, migration, and metastasis [5, 10, 11, 15, 16]. Furthermore, pH$_i$ dynamics regulate follicle stem cell differentiation in the *Drosophila* ovary and in mouse embryonic stem cells. [17]. During *Drosophila* oogenesis, stage-specific changes in pH$_i$ regulate follicle cell development by modifying the cytoskeleton and cell polarity [18, 19].

Few, if any studies have addressed the real-time response of intracellular pH to CO$_2$ in whole intact *Drosophila*. The duration and magnitude of large perturbations in pH could severely impact pH-sensitive processes such as cytoskeletal organization and morphogenesis and could be factors in whether, or how quickly, the system can recover from such perturbations.

Here, we sought to characterize the nature of the pH response to elevated CO$_2$ in living, intact, adult *Drosophila* using a pH sensor (tpHusion) and imaging directly through the abdominal cuticle. tpHusion is a genetically encoded, ubiquitously expressed pH sensor [20] consisting of a tubulin promoter, two fluorophores—pH-sensitive supereclliptic pHluorin and pH-insensitive FusionRed—and a short *HRas* sequence to tether the sensor to the intracellular membrane (Fig 1A). Whereas the intensity of pHluorin varies positively with the pH, FusionRed is insensitive. A change in the ratio of pHluorin-to-FusionRed intensities indicates a change in intracellular pH (Fig 1B). We imaged pHluorin and FusionRed intensities in the ovarian follicular epithelium directly through the ventral abdominal cuticle (Fig 1D, 1D' and 1D") using "Bellymount" (Fig 1C), a noninvasive method for imaging live, intact *Drosophila* [21].

## Results

We exposed control (*y w*;; *tpHusion*) and *Idgf-null* (*w*[1118] *Idgf*[4Δ]; *Idgf*[(1Δ dsRed, 2–3Δ, 6Δ, 5Δ)]; *tpHusion*) flies to a one-minute pulse of 100% CO$_2$, followed by a recovery period. To sustain a CO$_2$ flow rate of ~3.5 l/m into the CO$_2$ chamber, we used a Flowbuddy™ with a manual on/off switch and controlled the timing using the elapsed time displayed by the Leica software during image acquisition (see Methods). We monitored the change in pH$_i$ (inferred from the ratio of pHluorin/FusionRed intensities) in the main-body follicle cells in mid-stage egg chambers (Fig 1D, 1D' and 1D"). The pHluorin fluorescence intensity dropped sharply and nearly disappeared for both genotypes within 30 seconds while the FusionRed intensity remained relatively constant (examples in Fig 2A, 2A', 2B and 2B', S1 and S2 Videos). The ratios, normalized to 1.00 at the beginning of the CO$_2$ pulse, showed at least a two-fold decrease (Fig 2A" and 2B"). Fig 2C shows the normalized ratio plots for the control (n = 11) and *Idgf-null* (n = 11) flies, and Fig 2C' shows the averaged ratio profiles for each genotype. The initial response (0–2 minutes) was remarkably similar in both genotypes, but the response began to diverge during the recovery phase.

After removal of the CO$_2$, the pH returned to normal within a few minutes. The recovery time was measured starting at the time that the CO$_2$ was turned off (at 2 minutes) until the pHluorin/FusionRed ratio returned to a value of 1.0. The intracellular pH in *Idgf-null* flies recovered faster than in the control (3.2 minutes in *Idgf-null* vs. 4.7 minutes in control, *p* = 0.0029, Student's *t*-test) (Fig 2D). Note that the recovery time includes the time for the CO$_2$ to clear from the CO$_2$ chamber, so the start of normoxia is slightly delayed.

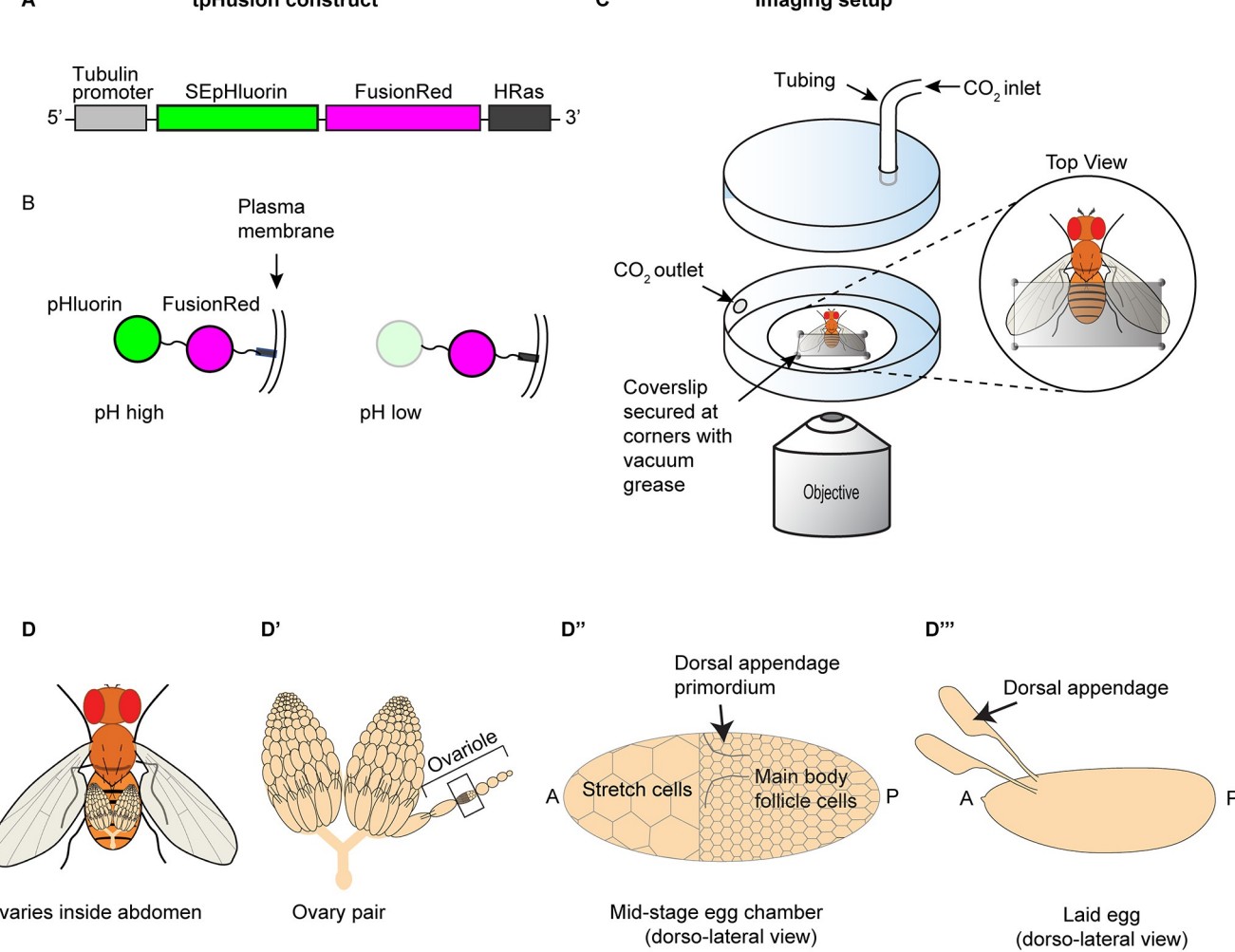

**Fig 1. pH sensor construct and live-imaging setup.** (A) The tpHusion construct consists of a fusion protein of supercliptic pHluorin (SEpHluorin) and FusionRed cloned under control of a tubulin promoter (alpha-Tub84B), which induces ubiquitous expression. An *HRas* sequence is included to tether the protein to the plasma membrane [20]. (B) Intracellular configuration of tpHusion. SEpHluorin fluoresces more brightly at high pH than at low pH. FusionRed is relatively insensitive to pH. (C) Imaging setup: a CO$_2$ chamber consists of a glass-bottom Petri dish with a CO$_2$ inlet and outlet. CO$_2$ flow rate was ~3.5 l/m (see Methods). The ventral abdomen of the living fly is glued to the glass-bottom coverslip with UV glue and slightly flattened using a small piece of cut coverslip secured with a dab of vacuum grease at each corner. (D) Schematic showing orientation of ovaries within a fly. (D') A pair of ovaries and an ovariole pulled out from an ovary. Egg chambers develop in assembly-line fashion from anterior to posterior along each ovariole. (D") Enlarged view of the egg chamber indicated by the rectangle in D'. Fluorescence intensity was measured within a subset of main body follicle cells, excluding dorsal appendage primordia (indicated by the two curved lines on the dorsal side of the egg chamber). The dorsal appendage primordia develop into tubes that form the dorsal appendages. (D''') Laid egg with dorsal appendages indicated (not to scale).

## Discussion

We found that pH$_i$ in *Drosophila* ovarian cells drops sharply, in just a few seconds, in response to a one-minute pulse of 100% CO$_2$, and then recovers gradually over several minutes after removal of the CO$_2$. Our findings give a real-time picture of how pH$_i$ responds in a living, intact animal, rather than in dissected or fixed samples. Dissected tissues are removed from their normal microenvironment inside the animal and can potentially be influenced by the pH of the buffer. The time it takes to dissect and fix tissues is problematic for capturing a rapid

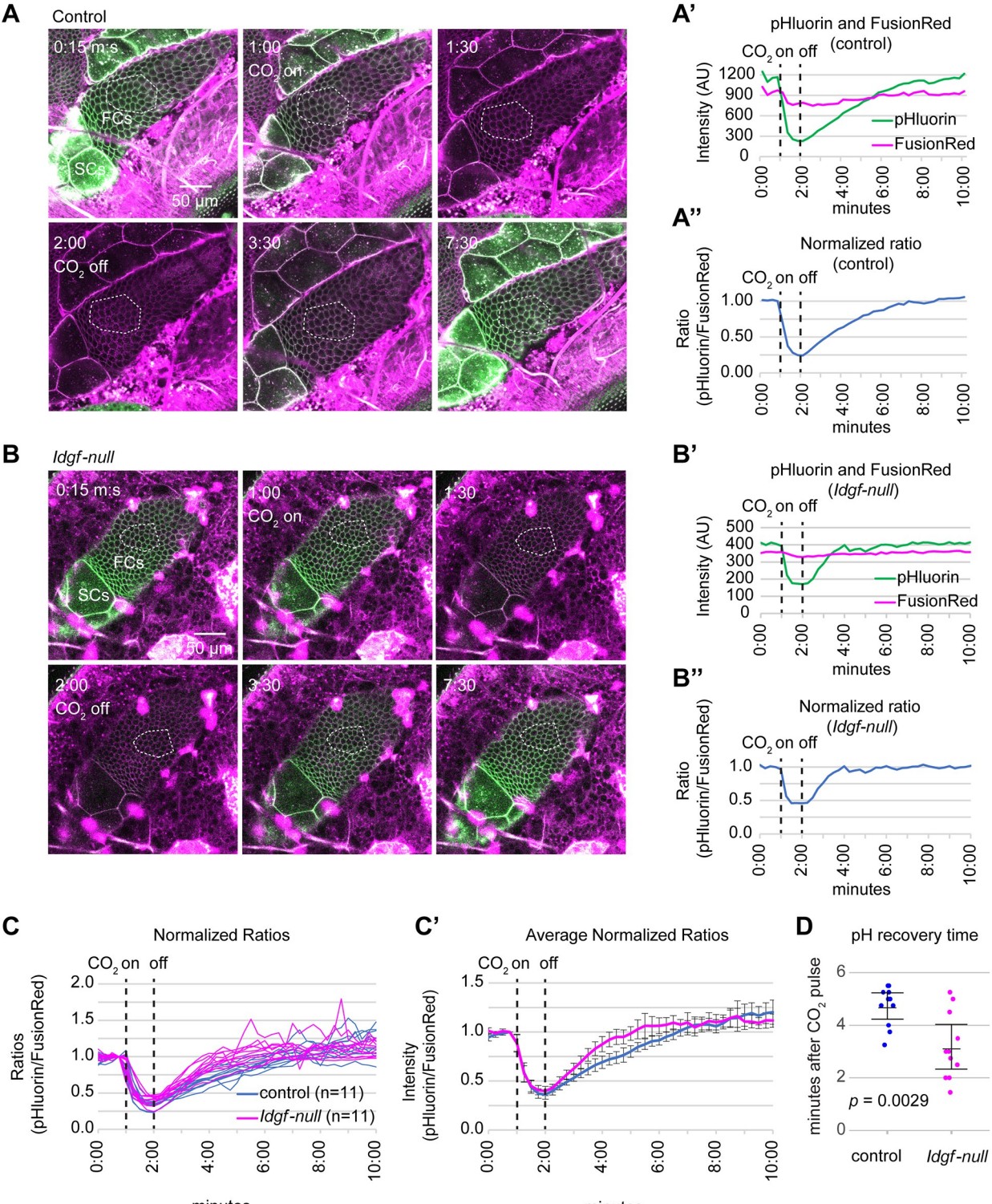

**Fig 2. pHluorin/FusionRed CO$_2$ response and recovery.** (A, B) Examples of still images extracted from videos (S1 and S2 Videos) of control (*y w;; tpHusion*) and *Idgf-null* (*w$^{1118}$ Idgf$^{4\Delta}$; Idgf$^{(1\Delta\ dsRed,\ 2-3\Delta,\ 6\Delta,\ 5\Delta)}$; tpHusion*) egg chambers in live, intact, adult *Drosophila*. Dotted lines indicate ROIs measured within main-body follicle cells. The brightness of the still images was increased for better visualization. Identical adjustments were applied to all images within a series. The videos were not adjusted. (A', B') 10-minute time lapse of pHluorin and FusionRed intensities from the same samples shown in panels A and B. (A", B") pHluorin/FusionRed intensity ratios normalized to 1.0 just prior to the 1-minute CO$_2$ pulse and showing the decrease and return to 1.0. Note: The images in panel B were scanned as 512x512-pixel images with an acquisition time of less than

five seconds. The images in panel A were scanned as 1024x1024-pixel so that at the same scan speed, the images took double the acquisition time of those in panel B. The longer acquisition captured a small decrease in the pH within the 1-minute frame immediately after the CO$_2$ was turned on. This decrease is evident in the second image in panel A and in the graphs in A' and A", showing a slight drop in the ratio in the frame starting at 1-minute. All of the wild-type videos (S1 and S3–S12 Videos) and *Idgf-null* videos S13–S20 Videos were scanned as 1024x1024 images. *Idgf-null* videos S2, S21 and S22 Videos were scanned as 512x512 images. (C) Normalized 10-minute time-lapse of pHluorin/FusionRed ratios for 11 control and 11 *Idgf-null* flies. Each line represents the profile for one fly. (C') Averaged normalized pHluorin/FusionRed ratios—averages of the profiles of the 11 control and 11 *Idgf-null* flies shown in panel C. Error bars represent 95% confidence limits. (D) Recovery time for pHluorin/FusionRed ratios. Each dot represents the amount of time between when the CO$_2$ was turned off to when the pHluorin/FusionRed ratio returned to 1.0. Error bars represent 95% confidence limits.

and dynamic pH response to CO$_2$ exposure, and the tissue pH can be influenced by the pH of the buffer or mounting media (personal observation).

Despite the ephemeral nature of the CO$_2$ pulse and the resulting drop in pH, the cytoskeletal and morphogenetic consequences are relatively long-term. The loss of cortical actin we previously observed in both wild-type and *Idgf-null* flies was apparent in embryos and in ovaries dissected and fixed 30 minutes after a single, one-minute, 100% CO$_2$ pulse. CO$_2$ exposure enhanced dorsal appendage defects in *Idgf-null* flies—dorsal appendages are eggshell structures produced by the follicular epithelium when two patches of cells form tubes and secrete eggshell proteins into the tube lumens. Thus, the dorsal appendages provide a readout for proper tube formation (Fig 1D" and 1D"'). The defects in *Idgf-null* dorsal appendages peaked at 8–10 hours after the CO$_2$ pulse and returned to the basal level by 27 hours [3]. The wild-type flies developed normally despite the loss of cortical actin apparent at 30 minutes post CO$_2$ exposure.

Considering the sensitivity of the cytoskeleton to changes in pH$_i$, we hypothesize that these cytoskeletal and morphogenetic defects are a direct result of the pH$_i$ perturbation, and the *Idgf-null* flies are less robust to perturbations in pH$_i$ than wild-type flies. Evidently a slower recovery to pH homeostasis in control versus *Idgf-null* ovarian cells is either not a factor in determining normal dorsal appendage development or could actually be beneficial. We propose that, rather than buffering against the pH$_i$ change itself, Idgfs protect against pH$_i$-induced cytoskeletal modifications by indirectly modulating the cytoskeleton, which is sensitive to pH. Future studies exploring the nature of the pH-induced cytoskeletal changes (e.g., monitoring endogenously expressed fluorescent markers for key proteins such as Idgfs, actin, and actin-modifying proteins) could shed light on this unexpected result.

The rapid drop in pH$_i$ was nearly identical in the two genotypes, yet the pH$_i$ in ovarian cells recovered faster in *Idgf-null* flies than in control flies. The rapid drop in pH$_i$ likely results from passive diffusion of dissolved CO$_2$ down a concentration gradient across the cell membranes and a reaction with water to form carbonic acid, which dissociates into hydrogen ions (H$^+$) and bicarbonate ions (HCO$_3^-$). The relatively slower return to homeostasis after removal of the CO$_2$ depends on transport of ions across the cell membranes and is regulated by carbonic anhydrases and several membrane pumps and transporters [reviewed in 5, 17–19] as well as pH-gated channels [22, 23]. Changes in expression or activity of these proteins could impact the rate at which pH returns to a pre-CO$_2$ level, but whether Idgfs influence these activities is unknown. Future studies defining the molecular mechanisms of Idgfs should give insight into this process and explain why pH$_i$ recovers more rapidly in flies lacking Idgfs.

We previously tested how different CO$_2$ exposure regimes affected dorsal appendage development, including a single 1-minute pulse of 100% CO$_2$, which increased defects in *Idgf-null* eggs but not in control eggs; a 1-minute pulse of 100% CO$_2$ every 12 hours for 3.5 days, which increased defects in *Idgf-null* eggs but not in control eggs; and continuous 20% CO$_2$ for up to eight days, which produced no increase in defects in either genotype [3]. Note that CO$_2$ makes

up about 0.04% of the ambient atmosphere [24]. Similar experiments monitoring the pH response to different exposure regimes will determine whether the cells can manage to maintain pH homeostasis under lower (e.g., $\leq 20\%$) $CO_2$ concentrations and for how long, whether the pH recovery profile is sensitive to the length of the $CO_2$ stimulus, and whether prior exposure to $CO_2$ at different concentrations alters the sensitivity of the pH response. A study in mice showed that, after an initial decline in arterial pH, after three days of exposure to 10% $CO_2$ pH returned to normal as a result of an increase in $HCO_3^-$ due to renal compensation [25]. Could Malpighian tubules, the *Drosophila* renal organ in insects, play a similar adaptive role in maintaining pH at lower concentrations of $CO_2$? Our previous finding that dorsal appendage defects are enhanced in *Idgf-null* flies when exposed to a single 1-minute pulse of 100% $CO_2$ but not after continuous exposure of 20% $CO_2$ for several days suggests the possibility that some sort of adaptation could be occurring at lower $CO_2$ concentrations.

*Drosophila* sense $CO_2$ through co-expression of gustatory receptors (Gr21a and Gr63a) in the antennae; these receptors activate specific neurons that affect behavior, i.e., attraction to $CO_2$ at low concentrations and avoidance at high concentrations [26]. Also, flies sense and avoid acidity through stimulation of different antennal nerves that express Ionotropic Receptor IR64a. $CO_2$ dissolved in the fluid inside the antennae can form carbonic acid and activate these acid-sensitive neurons [27]. It would be interesting to test whether the pH response is in any way altered in flies lacking these receptors. We predict, however, that the follicle-cell response would not differ in flies lacking these receptors. Such a response would depend on some sort of neural stimulus that would alter follicle cell physiology. Even given that the flies see $CO_2$ for a whole minute, this scenario seems unlikely. Our hypothesis is that $CO_2$ simply enters through the spiracles and into the tracheal system, then diffuses into the hemolymph and across cell membranes of the muscle sheath and follicle cells of the ovary. Thus, $CO_2$ affects pH by entering cells passively rather than by stimulating a G-protein coupled receptor.

One of the challenges in this study was that the proximity of specific egg chambers to the coverslip was variable from sample to sample. Within samples, peristaltic movements of the muscles surrounding the egg chambers produced sometimes dramatic movements of the egg chambers. This movement required adjustment to the imaging parameters (gain, laser intensity, and pinhole) from sample to sample. Longer imaging times for some samples with higher laser intensity could impact the results due to photobleaching after 8–10 minutes. We found that if we could image the samples with low intensities and lower resolution with faster scan times, we were able to avoid significant photobleaching during a 10-minute interval. Variation in the imaging parameters did not affect the results because each pHluorin/FusionRed ratio was normalized to the frame just prior to the beginning of the $CO_2$ pulse so that the beginning ratio just before the $CO_2$ pulse had a value of 1.0 for each profile. Therefore, all ratio curves were directly comparable.

Another limitation of this study is that because we used living, breathing, animals, we could not create a calibration curve to relate our pHluorin/FusionRed ratios to a known $pH_i$ within the animal. Our objective, however, was to monitor the change in $pH_i$, rather than the absolute $pH_i$.

Understanding pH dynamics in a tractable model organism such as *Drosophila* can provide a better understanding of pH alterations in human pathologies such as cancer [11], neurodegenerative disorders [28], and several other diseases [29]. In this study we have characterized the intracellular pH dynamics in response to a $CO_2$ pulse in a live intact animal using the Bellymount method and a genetically encoded, ubiquitously expressed pH sensor. These tools provide opportunities to explore the effects of pH dysregulation in fly ovaries as well as a variety of other tissues [20, 21].

## Methods

### Fly stocks

The control *y w;; tpHusion* stock expresses the transgene ubiquitously from a tubulin promoter [20] (gift from Hugo Stocker). The *y w;; tpHusion* stock was crossed into the *w$^{1118}$ Idgf$^{4\Delta}$; Idgf$^{(1\Delta\ dsRed,\ 2-3\Delta,\ 6\Delta,\ 5\Delta)}$* background to create a *w$^{1118}$ Idgf$^{4\Delta}$; Idgf$^{(1\Delta\ dsRed,\ 2-3\Delta,\ 6\Delta,\ 5\Delta)}$; tpHusion* stock. Flies were maintained on standard yeast, cornmeal, and molasses food at 25˚C.

### Imaging setup

Female flies fattened on wet yeast paste for two days prior to experimentation were immobilized by placing flies in empty vials and chilling them on ice for 30–60 minutes before mounting them. Each fly was secured to a glass-bottom Petri dish (cat. no. PG35G-1.5-14-C) using a small dab of UV glue (Bondic Pro UV Resin Kit, Amazon) while carefully avoiding obstructing the spiracles with glue. Elmer's Liquid School Glue is another option that doesn't dry as quickly but it is so easy to remove from the fly without injury that the fly can be returned to its vial and imaged again hours or days later. The fly was oriented with the ventral side toward the glass bottom and gently flattened using a piece of cover slip with a dab of vacuum grease (Dow Corning DC-HI-VAC-5.3OZ Silicone-Based High Vacuum Grease) at each corner to secure it to the bottom of the Petri dish.

### Fabrication of CO$_2$ chamber

The CO$_2$ chamber was made by drilling a hole in the top of the Petri dish cover with a fine-gauge hand drill (57 Pieces Hand Drill Bits Set, Amazon) so that a piece of tubing could be inserted to allow an inlet for the CO$_2$. A second hole was drilled in the side of the Petri dish bottom for the CO$_2$ outlet. CO$_2$ was supplied from a CO$_2$ tank connected to a FlowBuddy™ (The Flowbuddy™, Genesee Scientific, cat. no. 59-122B), for finer regulation of the CO$_2$ flow (approximately 3.5 liters/minute). Timing was controlled using the manual on/off switch on the FlowBuddy™ and the elapsed time displayed by the Leica software during image acquisition.

### Microscopy

Images were acquired on a Leica SP8X LSM. Video images were acquired as 12-bit and either 1024x1024 resolution acquired every 15 seconds (S1 and S3–S20 Videos) or 512x512 (S2, S21 and S22 Videos) acquired every 5 seconds. Images were converted to AVI files with 1 frame every 15 seconds. CO$_2$ was turned on at one minute, turned off at two minutes, and images were acquired for a total of 10 minutes. Due to variations in the proximity of particular egg chambers to the cuticle, imaging parameters (gain, laser intensity, and pinhole) were optimized for each specimen (see S1 Data). The pHluorin channel was acquired using a 475-nm laser line and the FusionRed channel was acquired using a 580-nm laser line.

### Image analysis and quantification

For each video frame, average pHluorin and FusionRed intensities were measured within a Region of Interest (ROI) located in the main-body follicle cells using ImageJ (Figs 1D", 2A and 2B and S1–S22 Videos). To generate profile plots, pHluorin/FusionRed intensity ratios were calculated in Excel at 15 second intervals for a total of 10 minutes. To allow comparison of ratio plots, each ratio was normalized to the frame just prior to the beginning of the CO$_2$ pulse so that the beginning ratio just before the CO$_2$ pulse had a value of 1.0 for each profile. The recovery time was the number of minutes from the time CO$_2$ was turned off to when the ratio returned to 1.0.

### Statistical analysis

Mean and confidence limits for Fig 2C' were calculated in Excel Version 2311. The mean, confidence limits, *p* value, and chart in Fig 2D were generated with R Version 4.0.2 using the code in S1 Text and the recovery times in S2 Data as input.

## Supporting information

**S1 Video. AVI video of control (*y w;; tpHusion*) egg chamber in Fig 2A.** Merged video of SEpHluorin (green) and FusionRed (magenta) fluorescence intensities in response to CO$_2$ exposure in live adult *Drosophila*. Yellow line indicates region in the main body follicle cells where average intensity was measured for each channel separately. Measurements are recorded in S1 Data.
(AVI)

**S2 Video. AVI video of *Idgf-null* (*w$^{1118}$ Idgf$^{4\Delta}$; Idgf$^{(1\Delta\ dsRed,\ 2-3\Delta,\ 6\Delta,\ 5\Delta)}$; tpHusion*) egg chamber in Fig 2B.** Merged video of SEpHluorin (green) and FusionRed (magenta) fluorescence intensities in response to CO$_2$ exposure in live adult *Drosophila*. White line indicates region in the main body follicle cells where average intensity was measured for each channel separately. Measurements are recorded in S1 Data.
(AVI)

**S3 Video. AVI video of control (*yw;; tpHusion*) egg chamber.** Merged video of SEpHluorin (green) and FusionRed (magenta) fluorescence intensities in response to CO$_2$ exposure in live adult *Drosophila*. Yellow line indicates region in the main body follicle cells where average intensity was measured for each channel separately. Measurements are recorded in S1 Data.
(AVI)

**S4 Video. AVI video of control (*y w;; tpHusion*) egg chamber.** Merged video of SEpHluorin (green) and FusionRed (magenta) fluorescence intensities in response to CO$_2$ exposure in live adult *Drosophila*. Yellow line indicates region in the main body follicle cells where average intensity was measured for each channel separately. Measurements are recorded in S1 Data.
(AVI)

**S5 Video. AVI video of control (*y w;; tpHusion*) egg chamber.** Merged video of SEpHluorin (green) and FusionRed (magenta) fluorescence intensities in response to CO$_2$ exposure in live adult *Drosophila*. Yellow line indicates region in the main body follicle cells where average intensity was measured for each channel separately. Measurements are recorded in S1 Data.
(AVI)

**S6 Video. AVI video of control (*y w;; tpHusion*) egg chamber.** Merged video of SEpHluorin (green) and FusionRed (magenta) fluorescence intensities in response to CO$_2$ exposure in live adult *Drosophila*. Yellow line indicates region in the main body follicle cells where average intensity was measured for each channel separately. Measurements are recorded in S1 Data.
(AVI)

**S7 Video. AVI video of control (*y w;; tpHusion*) egg chamber.** Merged video of SEpHluorin (green) and FusionRed (magenta) fluorescence intensities in response to CO$_2$ exposure in live adult *Drosophila*. Yellow line indicates region in the main body follicle cells where average intensity was measured for each channel separately. Measurements are recorded in S1 Data.
(AVI)

**S8 Video. AVI video of control (*y w;; tpHusion*) egg chamber.** Merged video of SEpHluorin (green) and FusionRed (magenta) fluorescence intensities in response to CO$_2$ exposure in live

adult *Drosophila*. Yellow line indicates region in the main body follicle cells where average intensity was measured for each channel separately. Measurements are recorded in S1 Data. (AVI)

**S9 Video. AVI video of control (*y w;; tpHusion*) egg chamber.** Merged video of SEpHluorin (green) and FusionRed (magenta) fluorescence intensities in response to CO$_2$ exposure in live adult *Drosophila*. Yellow line indicates region in the main body follicle cells where average intensity was measured for each channel separately. Measurements are recorded in S1 Data. (AVI)

**S10 Video. AVI video of control (*y w;; tpHusion*) egg chamber.** Merged video of SEpHluorin (green) and FusionRed (magenta) fluorescence intensities in response to CO$_2$ exposure in live adult *Drosophila*. Yellow line indicates region in the main body follicle cells where average intensity was measured for each channel separately. Measurements are recorded in S1 Data. (AVI)

**S11 Video. AVI video of control (*y w;; tpHusion*) egg chamber.** Merged video of SEpHluorin (green) and FusionRed (magenta) fluorescence intensities in response to CO$_2$ exposure in live adult *Drosophila*. Yellow line indicates region in the main body follicle cells where average intensity was measured for each channel separately. Measurements are recorded in S1 Data. (AVI)

**S12 Video. AVI video of control (*y w;; tpHusion*) egg chamber.** Merged video of SEpHluorin (green) and FusionRed (magenta) fluorescence intensities in response to CO$_2$ exposure in live adult *Drosophila*. Yellow line indicates region in the main body follicle cells where average intensity was measured for each channel separately. Measurements are recorded in S1 Data. (AVI)

**S13 Video. AVI video of *Idgf-null* (*w$^{1118}$ Idgf$^{4\Delta}$; Idgf$^{(1\Delta\ dsRed,\ 2-3\Delta,\ 6\Delta,\ 5\Delta)}$; tpHusion*) egg chamber.** Merged video of SEpHluorin (green) and FusionRed (magenta) fluorescence intensities in response to CO$_2$ exposure in live adult *Drosophila*. Yellow line indicates region in the main body follicle cells where average intensity was measured for each channel separately. Measurements are recorded in S1 Data. (AVI)

**S14 Video. AVI video of *Idgf-null* (*w$^{1118}$ Idgf$^{4\Delta}$; Idgf$^{(1\Delta\ dsRed,\ 2-3\Delta,\ 6\Delta,\ 5\Delta)}$; tpHusion*) egg chamber.** Merged video of SEpHluorin (green) and FusionRed (magenta) fluorescence intensities in response to CO$_2$ exposure in live adult *Drosophila*. Yellow line indicates region in the main body follicle cells where average intensity was measured for each channel separately. Measurements are recorded in S1 Data. (AVI)

**S15 Video. AVI video of *Idgf-null* (*w$^{1118}$ Idgf$^{4\Delta}$; Idgf$^{(1\Delta\ dsRed,\ 2-3\Delta,\ 6\Delta,\ 5\Delta)}$; tpHusion*) egg chamber.** Merged video of SEpHluorin (green) and FusionRed (magenta) fluorescence intensities in response to CO$_2$ exposure in live adult *Drosophila*. Yellow line indicates region in the main body follicle cells where average intensity was measured for each channel separately. Measurements are recorded in S1 Data. (AVI)

**S16 Video. AVI video of *Idgf-null* (*w$^{1118}$ Idgf$^{4\Delta}$; Idgf$^{(1\Delta\ dsRed,\ 2-3\Delta,\ 6\Delta,\ 5\Delta)}$; tpHusion*) egg chamber.** Merged video of SEpHluorin (green) and FusionRed (magenta) fluorescence intensities in response to CO$_2$ exposure in live adult *Drosophila*. Yellow line indicates region in the

main body follicle cells where average intensity was measured for each channel separately. Measurements are recorded in S1 Data.

(AVI)

**S17 Video. AVI video of *Idgf-null* (*w$^{1118}$ Idgf$^{4Δ}$; Idgf$^{(1Δ\ dsRed,\ 2-3Δ,\ 6Δ,\ 5Δ)}$; tpHusion*) egg chamber.** Merged video of SEpHluorin (green) and FusionRed (magenta) fluorescence intensities in response to CO$_2$ exposure in live adult *Drosophila*. Yellow line indicates region in the main body follicle cells where average intensity was measured for each channel separately. Measurements are recorded in S1 Data.

(AVI)

**S18 Video. AVI video of *Idgf-null* (*w$^{1118}$ Idgf$^{4Δ}$; Idgf$^{(1Δ\ dsRed,\ 2-3Δ,\ 6Δ,\ 5Δ)}$; tpHusion*) egg chamber.** Merged video of SEpHluorin (green) and FusionRed (magenta) fluorescence intensities in response to CO$_2$ exposure in live adult *Drosophila*. Yellow line indicates region in the main body follicle cells where average intensity was measured for each channel separately. Measurements are recorded in S1 Data.

(AVI)

**S19 Video. AVI video of *Idgf-null* (*w$^{1118}$ Idgf$^{4Δ}$; Idgf$^{(1Δ\ dsRed,\ 2-3Δ,\ 6Δ,\ 5Δ)}$; tpHusion*) egg chamber.** Merged video of SEpHluorin (green) and FusionRed (magenta) fluorescence intensities in response to CO$_2$ exposure in live adult *Drosophila*. Yellow line indicates region in the main body follicle cells where average intensity was measured for each channel separately. Measurements are recorded in S1 Data.

(AVI)

**S20 Video. AVI video of *Idgf-null* (*w$^{1118}$ Idgf$^{4Δ}$; Idgf$^{(1Δ\ dsRed,\ 2-3Δ,\ 6Δ,\ 5Δ)}$; tpHusion*) egg chamber.** Merged video of SEpHluorin (green) and FusionRed (magenta) fluorescence intensities in response to CO$_2$ exposure in live adult *Drosophila*. Yellow line indicates region in the main body follicle cells where average intensity was measured for each channel separately. Measurements are recorded in S1 Data.

(AVI)

**S21 Video. AVI video of *Idgf-null* (*w$^{1118}$ Idgf$^{4Δ}$; Idgf$^{(1Δ\ dsRed,\ 2-3Δ,\ 6Δ,\ 5Δ)}$; tpHusion*) egg chamber.** Merged video of SEpHluorin (green) and FusionRed (magenta) fluorescence intensities in response to CO$_2$ exposure in live adult *Drosophila*. Yellow line indicates region in the main body follicle cells where average intensity was measured for each channel separately. Measurements are recorded in S1 Data.

(AVI)

**S22 Video. AVI video of *Idgf-null* (*w$^{1118}$ Idgf$^{4Δ}$; Idgf$^{(1Δ\ dsRed,\ 2-3Δ,\ 6Δ,\ 5Δ)}$; tpHusion*) egg chamber.** Merged video of SEpHluorin (green) and FusionRed (magenta) fluorescence intensities in response to CO$_2$ exposure in live adult *Drosophila*. Yellow line indicates region in the main body follicle cells where average intensity was measured for each channel separately. Measurements are recorded in S1 Data.

(AVI)

**S1 Data. Numerical values measured from S1–S22 Videos and calculations supporting quantitative data presented in Fig 2A', 2A", 2B', 2B", 2C and 2C'.**

(XLSX)

**S2 Data. Recovery time after CO$_2$ exposure extracted from S1 Data in CSV format for input into R code in S1 Text.** Recovery time extends from when CO$_2$ is turned off (at 2

minutes) until the normalized pHluorin/FusionRed ratio returns to 1.0.
(CSV)

**S1 Text. R code for generating the chart and statistics for Fig 2D.**
(DOCX)

## Acknowledgments

The authors thank Hugo Stocker for the tpHusion fly stock and Todd Nystul for helpful advice on ratiometric imaging. They thank Nathaniel Peters at the University of Washington Keck Center for technical support and advice on imaging.

## Author Contributions

**Conceptualization:** Sandra G. Zimmerman, Celeste A. Berg.

**Data curation:** Sandra G. Zimmerman.

**Formal analysis:** Sandra G. Zimmerman.

**Funding acquisition:** Celeste A. Berg.

**Investigation:** Sandra G. Zimmerman.

**Methodology:** Sandra G. Zimmerman.

**Project administration:** Sandra G. Zimmerman, Celeste A. Berg.

**Resources:** Sandra G. Zimmerman, Celeste A. Berg.

**Software:** Sandra G. Zimmerman.

**Supervision:** Celeste A. Berg.

**Validation:** Sandra G. Zimmerman.

**Visualization:** Sandra G. Zimmerman.

**Writing – original draft:** Sandra G. Zimmerman.

**Writing – review & editing:** Sandra G. Zimmerman, Celeste A. Berg.

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
