## [Decision Letter · Decision Letter 0]

13 Feb 2024

PONE-D-23-42874CO_2_ exposure drives a rapid pH response in live adult *Drosophila*PLOS ONE

Dear Dr. Zimmerman,

Thank you for submitting your manuscript to PLOS ONE. After careful consideration, we feel that it has merit but does not fully meet PLOS ONE’s publication criteria as it currently stands. Therefore, we invite you to submit a revised version of the manuscript that addresses the points raised during the review process.

Both reviewers found the techniques presented in the manuscript were clearly described and scientifically useful. These reviewers were also able to offer some comments and suggestions that should improve the manuscript and make it useful for a wider audience.  Reviewer #1 became interested in the possibilities that the responses to CO2 may be due to sensory signaling from the known CO2 receptor.  Although I do not see the need to directly test this idea for this manuscript, it might be worth adding some text to discuss the role of CO2 perception, and how that may or  may not apply to the follicle cells.  This reviewer has requested some additional experiments to probe the nature of the pH responses detected, including time courses, recovery times, and dose responses.  Naturally, these could add significantly to our understanding of the responses of follicle cells to CO2, and would likely strengthen the paper's scientific impact, but I do not believe these are necessary for this publication, just something for consideration.  Please read through the other comments from this reviewer as they had several specific comments that a broader audience would likely also have, including additional experimental details on how the work was conducted.  Reviewer #2 had only two points that they numbered and clearly detailed.  I believe you can readily answer these points in the methods 1), and in the discussion 2), and I request you do so.    

We look forward to receiving your revised manuscript.

Kind regards,

Gregg Roman, PhD

Academic Editor

PLOS ONE

“(CB) National Institutes of Health, R01 GM079433, https://grantome.com/grant/NIH/R01-GM079433-01A2”

Reviewers' comments:

Reviewer's Responses to Questions

**Comments to the Author**

1. Is the manuscript technically sound, and do the data support the conclusions?

Reviewer #1: Yes

Reviewer #2: Yes

2. Has the statistical analysis been performed appropriately and rigorously? 

Reviewer #1: Yes

Reviewer #2: Yes

3. Have the authors made all data underlying the findings in their manuscript fully available?

Reviewer #1: Yes

Reviewer #2: Yes

4. Is the manuscript presented in an intelligible fashion and written in standard English?

Reviewer #1: Yes

Reviewer #2: Yes

5. Review Comments to the Author

Reviewer #1: In this manuscript, Zimmerman et.al. demonstrates the feasibility of monitoring pH in live adult Drosophila. They performed imaging experiments in living, intact flies and monitored pH changes following CO2 exposure. I have enclosed my comments.

• Is the recovery time following CO2 exposure determined by the length of the stimulus? It would be nice if authors carried out similar experiments with two different stimulus lengths. Comparing a trial shorter than one minute to a trial longer than one minute would convincingly establish this.

• Does prior exposure to CO2 at different concentrations (low, medium, and high) alter the sensitivity of the pH response following CO2 exposure? In other words, does sensory adaptation play a role in modifying the sensitivity of the response?

• I noticed that 100% CO2 has been used for imaging. I request the authors show some dose response by varying the concentration of the CO2 pulse.

• I request that the authors provide details on how the duration of the stimulus was precisely maintained in the experiment. What flow rate was used for the experiment?

• Although I believe that the change in pH is caused by CO2 exposure, I request authors repeat experiments with CO2 receptor mutant flies.

• Please explain the mechanism that determines the recovery phase after CO2 exposure. How does that differ between the control and the idgf mutant flies? Add comments on whether altered recovery phase is linked to any physiological consequence.

• I request authors to comment on the feasibility of using this pH sensor in other tissues.

Reviewer #2: In the manuscript by Zimmerman and Berg, the authors leverage the ubiquitous expression of a previously developed pH biosensor known as tpHusion, to examine whether CO2 exposure affects the intracellular pH of ovarian follicle cells within a developing egg chamber of a live adult Drosophila female. The motivation appears to be a previous finding from this lab that CO2 exposure enhanced developmental defects caused by mutations in the Imaginal disc grow factors (idgfs). To begin to assay how CO2 exposure might affect living cells, they first developed a CO2 administration chamber that houses a live fly in such a manner that they can image the response in adult ovaries through the female abdominal cuticle. They then apply a CO2 pulse and examine the tpHusion response. They detected a very rapid decrease in intracellular pH in the ovarian follicle cells, which after flushing the CO2 from the chamber, then returns to baseline with a relatively slow kinetic profile. Using this methodology, the authors then asked whether mutants in the idgfs alter the intracellular pH of the follicle cells in any way. They find that in idgf mutants, the kinetics of intracellular pH recovery are faster than wild type, thus validating that the methodology and tpHusion construct can be used as a tool to help examine how changes in intracellular pH might contribute to CO2 induced developmental or physiological defects, and how mutants affect the CO2 induced pH changes.

This is a nice short study that demonstrates the utility of this new intracellular pH sensor for understanding how CO2 exposure alters intracellular pH which then can be correlated with other intracellular changes (cytoskeleton) that might contribute to Co2 induced alterations in development fertility behavior etc. The data is robust and appropriate statistical analysis has been applied. I only have two minor comments.

1) It would be useful for the reader to know a little more about the tpHusion expression line without having to go to the original paper. I thought it might have been Gal4 driven but this is a transgene reporter expressed “ubiquitously” using the tubulin promoter. Perhaps in the “Fly Stocks” paragraph on the authors might say “The control y,w;;tpHusion stock expresses the transgene ubiquitously from a tubulin promoter (ref 20).

2) Although it is not the primary point of the paper, as mentioned above, it appears that one motivation for the work was to see if alterations in intracellular pH response might be the mechanism by which idfg loss enhanced sensitivity to CO2. The authors refer to work which has previously demonstrated that components of cytoskeleton are pH sensitive, and this sensitivity correlates with morphogenic defects associated with CO2 exposure. However, the mutant idgf flies actually recover more quickly than wildtype, not more slowly as one might expect if the role of idgfs was to buffer against CO2 induced pH changes. In my view the authors should mention this conundrum, or perhaps I am missing something.

6. PLOS authors have the option to publish the peer review history of their article (what does this mean?). If published, this will include your full peer review and any attached files.

Reviewer #1: **Yes: **Tuhin Subhra Chakraborty

Reviewer #2: No

---

## [Author Response · Author response to Decision Letter 0]

19 Mar 2024

We thank the reviewers for their thorough analyses, insightful comments, and helpful suggestions. We are grateful for their help in improving our paper. 

Below we state the reviewer’s concern, describe how we have addressed each issue, and provide the revised text in quotation marks.

Editors comments: 

Both reviewers found the techniques presented in the manuscript were clearly described and scientifically useful. These reviewers were also able to offer some comments and suggestions that should improve the manuscript and make it useful for a wider audience. 

Reviewer #1 became interested in the possibilities that the responses to CO2 may be due to sensory signaling from the known CO2 receptor. Although I do not see the need to directly test this idea for this manuscript, it might be worth adding some text to discuss the role of CO2 perception, and how that may or may not apply to the follicle cells. This reviewer has requested some additional experiments to probe the nature of the pH responses detected, including time courses, recovery times, and dose responses. Naturally, these could add significantly to our understanding of the responses of follicle cells to CO2, and would likely strengthen the paper's scientific impact, but I do not believe these are necessary for this publication, just something for consideration. Please read through the other comments from this reviewer as they had several specific comments that a broader audience would likely also have, including additional experimental details on how the work was conducted. 

Reviewer #2 had only two points that they numbered and clearly detailed. I believe you can readily answer these points in the methods 1), and in the discussion 2), and I request you do so. 

Our response to reviewer comments

Reviewer #1:

1. Comment 1: Is the recovery time following CO2 exposure determined by the length of the stimulus? It would be nice if authors carried out similar experiments with two different stimulus lengths. Comparing a trial shorter than one minute to a trial longer than one minute would convincingly establish this.

Response: We did not perform this experiment but we added the following text in paragraph 5 of the discussion. We discussed the requested experiments as part of a section on future studies.

“We previously tested how different CO2 exposure regimes affected dorsal appendage development, including a single 1-minute pulse of 100% CO2, which increased defects in Idgf-null eggs but not in control eggs; a 1-minute pulse of 100% CO2 every 12 hours for 3.5 days, which increased defects in Idgf-null eggs but not in control eggs; and continuous 20% CO2 for up to eight days, which produced no increase in defects in either genotype [3]. Note that CO2 makes up about 0.04% of the ambient atmosphere [26]. Similar experiments monitoring the pH response to different exposure regimes will determine whether the cells can manage to maintain pH homeostasis under lower (e.g., ≤ 20%) CO2 concentrations and for how long, whether the pH recovery profile is sensitive to the length of the CO2 stimulus, and whether prior exposure to CO2 at different concentrations alters the sensitivity of the pH response. A study in mice showed that, after an initial decline in arterial pH, after three days of exposure to 10% CO2 pH returned to normal as a result of an increase in HCO3- due to renal compensation [27]. Could Malpighian tubules, the Drosophila renal organ in insects, play a similar adaptive role in maintaining pH at lower concentrations of CO2? Our previous finding that dorsal appendage defects are enhanced in Idgf-null flies when exposed to a single 1-minute pulse of 100% CO2 but not after continuous exposure of 20% CO2 for several days suggests the possibility that some sort of adaptation could be occurring at lower CO2 concentrations.”

2. Comment 2: Does prior exposure to CO2 at different concentrations (low, medium, and high) alter the sensitivity of the pH response following CO2 exposure? In other words, does sensory adaptation play a role in modifying the sensitivity of the response?

Response: This idea is really interesting, since it invokes the process that occurs normally downstream of G-protein-coupled receptors, e.g., those involved in smell and taste. In our previous study in Reference 3, we tested the effect of different concentrations and exposure times on dorsal appendage development. We noted that dorsal appendage defects were not enhanced at 20% CO2 exposure, even after several days of constant 20% CO2, and this lack of enhancement could suggest adaptation. We predict, however, that these G-protein-coupled receptors are not involved. Our hypothesis is that CO2 simply enters through the spiracles and into the tracheal system, then diffuses into the hemolymph and across cell membranes of the muscle sheath and follicle cells of the ovary. Thus, CO2 affects pH by entering cells passively rather than by stimulating a G-protein coupled receptor. The flies used in these experiments were never exposed to CO2 prior to the experiments, and we did not test whether previous CO2 exposure at different concentrations would alter the pH response due to sensory adaptation. We added the following text to paragraph 5 of the discussion to address possible adaptation:

“A study in mice showed that, after an initial decline in arterial pH after three days of exposure to 10% CO2, pH returned to normal as a result of an increase in HCO3- due to renal compensation [27]. Could Malpighian tubules, the Drosophila renal organ in insects, play a similar adaptive role in maintaining pH at lower concentrations of CO2? Our previous finding that dorsal appendage defects are enhanced in Idgf-null flies when exposed to a single1-minute pulse of 100% CO2 but not after continuous exposure of 20% CO2 for several days suggests the possibility that some sort of adaptation could be occurring at lower CO2 concentrations.”

3. Comment 3: I noticed that 100% CO2 has been used for imaging. I request the authors show some dose response by varying the concentration of the CO2 pulse.

Response: Varying the CO2 concentration would require special tanks with the ratios needed for exposing flies to precise concentrations of CO2. We did not perform this experiment but we added text to paragraph 5 of the discussion to address it as a possible future study (see revised text in the response to Comment 1). 

4. Comment 4: I request that the authors provide details on how the duration of the stimulus was precisely maintained in the experiment. What flow rate was used for the experiment?

Response: The flow rate and how it was controlled is described in the Methods section under “Fabrication of CO2 chamber”. 

We added a sentence explaining how the timing was controlled: 

“CO2 was supplied from a CO2 tank connected to a FlowBuddyTM (The FlowbuddyTM, Genesee Scientific, cat. no. 59-122B), for finer regulation of the CO2 flow (approximately 3.5 liters/minute). Timing was controlled using the manual on/off switch on the FlowBuddyTM and the elapsed time displayed by the Leica software during image acquisition.”

We added a description in the Results section of how flow rate and timing were controlled:

“To sustain a CO2 flow rate or ~3.5 l/m into the CO2 chamber, we used a FlowbuddyTM with a manual on/off switch and controlled the timing using the elapsed time displayed by the Leica software during image acquisition (see Methods).”

5. Comment 5: Although I believe that the change in pH is caused by CO2 exposure, I request authors repeat experiments with CO2 receptor mutant flies.

Response: This is an interesting question and we added text discussing CO2- and acid-sensing receptors and proposed future experiments using flies lacking these receptors. We added the following text to the discussion as paragraph 6 of the discussion:

“Drosophila sense CO2 through co-expression of gustatory receptors (Gr21a and Gr63a) in the antennae; these receptors activate specific neurons that affect behavior, i.e., attraction to CO2 at low concentrations and avoidance at high concentrations [28]. Also, flies sense and avoid acidity through stimulation of different antennal nerves that express Ionotropic Receptor IR64a. CO2 dissolved in the fluid inside the antennae can form carbonic acid and activate these acid-sensitive neurons [29]. It would be interesting to test whether the pH response is in any way altered in flies lacking these receptors. We predict, however, that the follicle cell response would not differ in flies lacking these receptors. Such a response would depend on some sort of neural stimulus that would alter follicle cell physiology. Even given that the flies see CO2 for a whole minute, this scenario seems unlikely. Our hypothesis is that CO2 simply enters through the spiracles and into the tracheal system, then diffuses into the hemolymph and across cell membranes of the muscle sheath and follicle cells of the ovary. Thus, CO2 affects pH by entering cells passively rather than by stimulating a G-protein coupled receptor.” 

6. Comment 6: Please explain the mechanism that determines the recovery phase after CO2 exposure. How does that differ between the control and the Idgf mutant flies? Add comments on whether altered recovery phase is linked to any physiological consequence.

Response: Thank you for suggesting this change. It is an important aspect of the process that needed clarifying. We have added text to paragraph 4 of the discussion to describe potential explanations for determining the pH recovery phase and why it may differ between the two genotypes:

“The rapid drop in pHi was nearly identical in the two genotypes, yet the pHi in ovarian cells recovered faster in Idgf-null flies than in control flies. The rapid drop in pHi likely results from passive diffusion of dissolved CO2 down a concentration gradient across the cell membranes and a reaction with water to form carbonic acid, which dissociates into hydrogen ions (H+) and bicarbonate ions (HCO3-). The relatively slower return to homeostasis after removal of the CO2, depends on transport of ions across the cell membranes and is regulated by carbonic anhydrases and several membrane pumps and transporters [reviewed in 5, 17-19] as well as pH-gated channels [24, 25]. Changes in expression or activity of these proteins could impact the rate at which pH returns to a pre-CO2 level, but whether Idgfs influence these activities is unknown. Future studies defining the molecular mechanisms of Idgfs should give insight into this process and explain why pHi recovers more rapidly in flies lacking Idgfs.” 

We commented on whether the different recovery profiles are linked to any physiological consequence, specifically, dorsal appendage development, based on our previous study [3]. We added the following revision to paragraphs 2 and 3 of the discussion:

“The defects in Idgf-null dorsal appendages peaked at 8 - 10 hours after the CO2 pulse and returned to the basal level by 27 hours [3]. The wild-type flies developed normally despite the loss of cortical actin apparent at 30 minutes post CO2 exposure. 

Considering the sensitivity of the cytoskeleton to changes in pHi, we hypothesize that these cytoskeletal and morphogenetic defects are a direct result of the pHi perturbation, and the Idgf-null flies are less robust to perturbations in pHi than wild-type flies. Evidently a slower recovery to pH homeostasis in control versus Idgf-null ovarian cells is either not a factor in determining normal dorsal appendage development or could actually be beneficial. We propose that, rather than buffering against the pHi change itself, Idgfs protect against pHi-induced cytoskeletal modifications by indirectly modulating the cytoskeleton, which is sensitive to pH. Future studies exploring the nature of the pH-induced cytoskeletal changes (e.g., monitoring endogenously expressed fluorescent markers for key proteins such as Idgfs, actin, and actin-modifying proteins) could shed light on this unexpected result."

7. Comment 7: I request authors to comment on the feasibility of using this pH sensor in other tissues.

Response: We added a sentence to the last paragraph of the discussion citing studies that have used this pH sensor and the live imaging procedure in other tissues:

“These tools provide opportunities to explore the effects of pH dysregulation in fly ovaries as well as a variety of other tissues [20, 21].”

Reviewer 2:

1. Comment 1: It would be useful for the reader to know a little more about the tpHusion expression line without having to go to the original paper. I thought it might have been Gal4 driven but this is a transgene reporter expressed “ubiquitously” using the tubulin promoter. Perhaps in the “Fly Stocks” paragraph on the authors might say “The control y,w;;tpHusion stock expresses the transgene ubiquitously from a tubulin promoter (ref 20).

Response: We added the sentence “The control y w; ;tpHusion stock expresses the transgene ubiquitously from a tubulin promoter.” to the methods. This feature is also described in the Introduction and we added a clarification in the legend for figure Figure 1 that it is ubiquitously expressed.

2. Comment 2: Although it is not the primary point of the paper, as mentioned above, it appears that one motivation for the work was to see if alterations in intracellular pH response might be the mechanism by which Idgf loss enhanced sensitivity to CO2. The authors refer to work which has previously demonstrated that components of cytoskeleton are pH sensitive, and this sensitivity correlates with morphogenic defects associated with CO2 exposure. However, the mutant Idgf flies actually recover more quickly than wildtype, not more slowly as one might expect if the role of idgfs was to buffer against CO2 induced pH changes. In my view the authors should mention this conundrum, or perhaps I am missing something.

Response: Thank you for pointing out this feature of the results that is not intuitive. We added text to the discussion to pose possible explanations for this “conundrum”. We have evidence from our previous study [Sustar AE, et al., Genetics. 2023;223(2)] that the presence of Idgfs protects against the damage to the actin cytoskeleton caused by exposure to CO2. We did not intend to suggest that the role of Idgfs is to buffer the pH but rather has a role in mitigating the cytoskeletal damage caused by a pH perturbation. We revised paragraphs 2, 3, and 4 as shown in the response to Reviewer 1, Comment 6.

In responding to reviewer comments, we have added the following references to the manuscript (note that the original Refs 22 and 23 are now Refs 28 and 29):

22. Feingold D, Starc T, O'Donnell MJ, Nilson L, Dent JA. The orphan pentameric ligand-gated ion channel pHCl-2 is gated by pH and regulates fluid secretion in Drosophila Malpighian tubules. J Exp Biol. 2016;219(Pt 17):2629-2638.

23. Schnizler K, Saeger B, Pfeffer C, Gerbaulet A, Ebbinghaus-Kintscher U, Methfessel C, et al. A novel chloride channel in Drosophila melanogaster is inhibited by protons. J Biol Chem. 2005;280(16):16254-16262.

24. Department of Energy. https://netl.doe.gov/coal/carbon-storage/faqs/carbon-dioxide-101

25. Gates KL, Howell HA, Nair A, Vohwinkel CU, Welch LC, Beitel GJ, et al. Hypercapnia impairs lung neutrophil function and increases mortality in murine pseudomonas pneumonia. Am J Respir Cell Mol Biol. 2013;49(5):821-828.

26. Kwon JY, Dahanukar A, Weiss LA, Carlson JR. The molecular basis of CO2 reception in Drosophila. Proc Natl Acad Sci U S A. 2007;104(9):3574-3578.

27. Ai M, Min S, Grosjean Y, Leblanc C, Bell R, Benton R, et al. Acid sensing by the Drosophila olfactory system. Nature. 2010;468(7324):691-695.

---

## [Editor Report · Decision Letter 1]

1 Apr 2024

CO_2_ exposure drives a rapid pH response in live adult *Drosophila*

PONE-D-23-42874R1

Dear Dr. Zimmerman,

We’re pleased to inform you that your manuscript has been judged scientifically suitable for publication and will be formally accepted for publication once it meets all outstanding technical requirements.

Kind regards,

Gregg Roman, PhD

Academic Editor

PLOS ONE
---

## [Editor Report · Acceptance letter]

3 Apr 2024

PONE-D-23-42874R1 

PLOS ONE

Dear Dr. Zimmerman, 

I'm pleased to inform you that your manuscript has been deemed suitable for publication in PLOS ONE. Congratulations! Your manuscript is now being handed over to our production team.

Kind regards, 

on behalf of

Dr Gregg Roman 

Academic Editor

PLOS ONE